# Highly Efficient Liquid-Phase Exfoliation of Layered Perovskite-like Titanates HLnTiO_4_ and H_2_Ln_2_Ti_3_O_10_ (Ln = La, Nd) into Nanosheets

**DOI:** 10.3390/nano13233052

**Published:** 2023-11-29

**Authors:** Sergei A. Kurnosenko, Iana A. Minich, Oleg I. Silyukov, Irina A. Zvereva

**Affiliations:** Department of Chemical Thermodynamics and Kinetics, Institute of Chemistry, Saint Petersburg State University, 199034 Saint Petersburg, Russia; st040572@student.spbu.ru (S.A.K.); yana.minich@spbu.ru (I.A.M.); irina.zvereva@spbu.ru (I.A.Z.)

**Keywords:** layered perovskite, titanate, exfoliation, nanosheets, suspension stability, amine, intercalation

## Abstract

Nanosheets of layered perovskite-like oxides attract researchers as building blocks for the creation of a wide range of demanded nanomaterials. However, Ruddlesden–Popper phases are difficult to separate into nanosheets quantitatively via the conventional liquid-phase exfoliation procedure in aqueous solutions of bulky organic bases. The present study has considered systematically a relatively novel and efficient approach to a high-yield preparation of concentrated suspensions of perovskite nanosheets. For this, the Ruddlesden–Popper titanates HLnTiO_4_ and H_2_Ln_2_Ti_3_O_10_ (Ln = La, Nd) have been intercalated by *n*-alkylamines with various chain lengths, exposed to sonication in aqueous tetrabutylammonium hydroxide (TBAOH) and centrifuged to separate the nanosheet-containing supernatant. The experiments included variations of a wide range of conditions, which allowed for the achievement of impressive nanosheet concentrations in suspensions up to 2.1 g/L and yields up to 95%. The latter were found to strongly depend on the length of intercalated *n*-alkylamines. Despite the less expanded interlayer space, the titanates modified with short-chain amines demonstrated a much higher completeness of liquid-phase exfoliation as compared to those with long-chain ones. It was also shown that the exfoliation efficiency depends more on the sample stirring time in the TBAOH solution than on the sonication duration. Analysis of the titanate nanosheets obtained by means of dynamic light scattering, electron and atomic force microscopy revealed their lateral sizes of 30–250 nm and thickness of 2–4 nm. The investigated exfoliation strategy appears to be convenient for the high-yield production of perovskite nanosheet-based materials for photocatalytic hydrogen production, environmental remediation and other applications.

## 1. Introduction

At the beginning of the XXI century, two-dimensional (2D) nanomaterials, including nanosheets, nanorods, nanowires and ultrathin films, have become the focus of attention of researchers and engineers due to their attractive mechanical, electrical, magnetic, optical and catalytic properties, which can differ markedly from those of three-dimensional (3D) bulk analogues on account of specific nanoscale effects [1]. Thanks to these features, 2D nanomaterials are highly demanded in such technologically complex areas as the creation of catalysts [2], photocatalysts [3], photoelectrochemical [4] and photovoltaic cells [5], supercapacitors [6], thermoelectrics [7], nanoelectronic devices [8], membranes [9], chemical sensors [10] and drug delivery systems [11].

Among nanostructured materials potentially applicable in energy production, biotechnology, ecology and related areas, ion-exchangeable layered perovskite-like oxides are of particular interest [12]. The latter are crystalline solids whose structure consists of negatively charged slabs with a thickness of n corner-shared perovskite octahedra, alternating regularly with ion-exchangeable interlayer spaces populated by cations. These oxides may be divided in two groups, namely the Dion–Jacobson phases and the Ruddlesden–Popper phases, following general formulae A′[A_n−1_B_n_O_3n+1_] and A′_2_[A_n−1_B_n_O_3n+1_], respectively (A′ = alkali metal, A = rare earth or transition metal, B = Ti, Zr, Nb, Ta, etc.) [13,14,15]. The superior physical–chemical properties of these compounds originate from the unique perovskite structure of the slabs and the propensity of their interlayer space for various reactions, such of ion exchange and intercalation [16,17,18,19]. Layered perovskites are well established as efficient catalysts [20,21], photocatalysts [22,23,24,25,26,27,28,29], ionic conductors [30,31,32,33,34,35,36,37,38] and luminescent materials [39,40,41,42,43,44], as well as precursors for the creation of composite [45,46,47,48,49,50,51,52] and hybrid inorganic–organic materials [53,54,55,56,57,58,59,60].

Exfoliation of layered perovskite-like oxides into 2D nanosheets is of high importance for their use in heterogeneous catalysis [61] and photocatalysis [62]. Being nanoscale objects, perovskite nanosheets have large specific surface areas and small sizes that provide an enhanced number of active sites, efficient reactant adsorption and reduced intensity of charge carrier recombination [62], which, in turn, explains their improved photocatalytic activity in comparison with that of their bulk counterparts. Moreover, perovskite nanosheets may be used as building blocks to obtain new layered materials via controllable self-assembly [63,64,65,66]. Since exfoliation and assembly processes are usually performed via soft chemistry techniques, final materials preserve crystal structure and, consequently, valuable properties of initial compounds.

Among the wide range of top-down and bottom-up methods for perovskite nanosheet preparation, the most common is liquid-phase exfoliation in aqueous media via intercalation of bulky cations (usually, tetrabutylammonium TBA^+^) between the slabs followed by sonication [67]. The intercalation results in significant expansion of the interlayer space and its intense hydration. Further ultrasonic treatment is supposed to generate a large number of dense bubble clusters whose powerful vibration and implosion produce severe energy boosts, mechanical shear stresses and turbulence, thus providing delamination for the layered structure [68]. Along with ultrasound, its initiation is also reported to be feasible via microwave treatment [69] and a shear-force assisted approach [70]. The intercalation-sonication method allows for the achievement of high exfoliation completeness for many layered compounds. However, cavitation may be accompanied by undesirable destruction of the perovskite slabs or formation of dissimilar particles with a too wide size distribution, which justifies the development of milder liquid-phase exfoliation techniques. For instance, some layered materials are capable of exfoliation by electrostatic repulsion, which can be caused by intercalation of a long-chain cationic amino acid followed by a medium alkalization to convert it to an anionic form repelled by the negatively charged slabs [71,72].

The layered perovskite-like titanates HLnTiO_4_ and H_2_Ln_2_Ti_3_O_10_ (Figure 1a) are protonated forms of the Ruddlesden–Popper phases ALnTiO_4_ and A_2_Ln_2_Ti_3_O_10_ (A = alkali metal, Ln = La or lanthanide), possessing the perovskite slab thickness of n = 1 and n = 3 titanium–oxygen octahedra, respectively [73,74]. Despite practically valuable catalytic [20,21] and photocatalytic properties [75,76,77,78,79,80,81,82], their liquid-phase exfoliation into nanosheets has not been sufficiently investigated. Although there are several reports mentioning the obtainment of their exfoliated forms [83,84,85,86], none of them covers a purposeful preparation of their highly concentrated suspensions, which would be of great interest for catalytic applications. This fact is apparently associated with the relatively lower reactivity of the Ruddlesden–Popper phases towards intercalation reactions, including those with TBAOH, as compared with the Dion–Jacobson ones whose liquid-phase exfoliation has been thoroughly studied [87,88,89,90,91,92,93,94,95,96].

The present paper is a systematic study of quantitative liquid-phase exfoliation of the titanates HLnTiO_4_ and H_2_Ln_2_Ti_3_O_10_ (Ln = La, Nd) under various conditions. Primarily, attention has been paid to the optimization of their exfoliation procedure in order to provide high nanosheet yields using *n*-alkylamine derivatives of the titanates with various chain lengths as precursors (Figure 1b), as well as the stability of the nanosheet suspensions and the morphology of the nanoparticles obtained.

## 2. Materials and Methods

Here and below, the following abbreviations are used: HLT and HNT for n = 1 titanates HLaTiO_4_ and HNdTiO_4_; HLT_3_ and HNT_3_ for n = 3 titanates H_2_La_2_Ti_3_O_10_ and H_2_Nd_2_Ti_3_O_10_; HLT×RNH_2_, HNT×RNH_2_, HLT_3_×RNH_2_ and HNT_3_×RNH_2_ for *n*-alkylamine derivatives (R = methyl Me, ethyl Et, *n*-propyl Pr, *n*-butyl Bu, *n*-hexyl Hx and *n*-octyl Oc).

### 2.1. Synthesis of Initial Protonated Titanates

Alkaline layered perovskite-like titanates NaLnTiO_4_ and K_2_Ln_2_Ti_3_O_10_ were prepared according to the conventional ceramic technique using calcined TiO_2_, Ln_2_O_3_ (Ln = La or Nd) and A_2_CO_3_ (A = Na or K) as starting compounds. The oxides were taken in stoichiometric amounts, sodium carbonate—with a 30% excess to compensate for the loss during calcination. All the reactants were placed into a grinding bowl with silicon nitride balls, flooded with *n*-heptane and ground in a Fritsch Pulverisette 7 planetary micro mill (Idar-Oberstein, Germany) at a rotation speed of 600 rpm using a program of 10 repetitions of 10 min each with 5 min intervals. The mixture obtained was dried and pelletized into ~2 g tablets at 50 bar using an Omec PI 88.00 hydraulic press (Certaldo, Italy). The tablets were placed into corundum crucibles with lids, kept at 825 °C (for NaLnTiO_4_) or 1100 °C (for K_2_Ln_2_Ti_3_O_10_) for 12 h in a Nabertherm L-011K2RN muffle furnace (Lilienthal, Germany) and, after cooling down, ground in an agate mortar.

Protonated titanates HLT (HNT) and HLT_3_ (HNT_3_) were obtained via the ion exchange. To prepare HLT (HNT), powders of NaLnTiO_4_ were treated with a 0.1 M hydrochloric acid solution taken at a rate of 200 mL per 1 g of the titanate at 25 °C for 1 d. In the case of HLT_3_ (HNT_3_) preparation, powders of K_2_Ln_2_Ti_3_O_10_ were kept in the atmosphere of humid air for 1 d. Hydrated alkaline titanates obtained in this way were treated with water at a rate of 200 mL per 1 g of the titanate for 1 h and then with a 0.1 M hydrochloric acid solution taken in the same ratio for 1 d. Then, solid phases were separated from the solution via centrifuging and dried in a desiccator over CaO for 2 d.

### 2.2. Synthesis of n-Alkylamine Derivatives

The conditions for the intercalation of *n*-alkylamines into the interlayer space of the protonated titanates were adopted from our previous reports [27,57]. Methylamine derivatives (R = Me) were synthesized directly from the protonated titanates. For this, 3 g of each titanate was thoroughly ground in an agate mortar, placed in a sealed glass tube containing 30 mL of 38% aqueous methylamine and stirred at 60 °C for 10 d. Synthesis of other amine derivatives (R = Et, Pr, Bu, Hx, Oc) was carried out using methylamine ones as precursors. In each case, 1 g of the methylamine derivative was placed in a sealed glass tube with 10 mL of a corresponding amine solution, after which the reaction mixture was stirred at the given temperature T for the predetermined time D. After cooling down, the solid products were filtered and rinsed with an appropriate volatile solvent to remove residual surface adsorbed amines. Detailed conditions for the intercalation of *n*-alkylamines are summarized in Table 1.

### 2.3. Optimization of the Exfoliation Procedure

Exfoliation of the titanates into nanosheets was investigated using physical, chemical and physical–chemical strategies; the latter was paid the most attention due to significantly higher efficiency. The physical method consisted in sonication of the layered precursor in water without TBAOH or other exfoliating agents. The chemical approach represented stirring the initial compound in an aqueous TBAOH solution for a given time without ultrasonic treatment. The combined physical–chemical strategy included sonication of the precursor in a TBAOH solution followed by stirring for a specified time, repeated sonication in the same mode and centrifuging (Figure 1b). 

The ultrasonic treatment was performed by a 200 W ultrasonic homogenizer Hielscher UP200St (Teltow, Germany) via placing the 7 nm sonotrode directly into a glass tube with the suspension. In all cases, the mixture obtained was centrifuged on an Elmi CM-6MT laboratory centrifuge (Riga, Latvia) to precipitate bulk non-exfoliated particles, and the final suspension of nanosheets was separated from the sediment by a glass pipette. The default exfoliation conditions, unless otherwise noted, were as follows: a precursor weight (for the *n*-alkylamine derivatives—in terms of their inorganic part): 30 mg, solution volume: 30 mL, TBAOH concentration: 0.004 M, sonication power P: 50% from the nominal value, single sonication duration: 5 min, stirring duration: 1 d, separation factor for centrifuging: F = 1000, centrifuging duration: 1 h. The precursor chosen and the aforementioned experimental parameters were varied in order to find the optimal exfoliation conditions that would provide the highest yield of nanosheets in the final suspensions (Table 2). In each series of experiments, only one parameter was varied, while the rest were taken by default.

Suspensions prepared via the exfoliation of ethylamine-intercalated titanates under the aforementioned default conditions (standard suspensions) were used for building spectrophotometric calibration plots, further in-depth investigation of nanosheets and their deposition on substrates.

### 2.4. Construction of Spectrophotometric Calibration Plots

Spectrophotometric calibrations were built for express measurement of nanosheet concentrations in the final suspensions obtained under various conditions. For this, the nanosheet concentrations in the standard suspensions were determined by inductively coupled plasma atomic emission spectroscopy after acid digestion. Then, spectra of a series of diluted suspensions were recorded, analytical wavelengths were selected (λ = 240 nm for HLT, λ = 225 nm for HNT, λ = 230 nm for HLT_3_ and HNT_3_) and the experimental dependence of the optical density on nanosheet concentration A_λ_ = A_λ_(c) was approximated by a linear function using the least-squares method.

### 2.5. Deposition of the Nanosheets on Substrates

To investigate the morphology of separate nanosheets without preliminary flocculation, they were deposited on silicon substrates via the self-assembly method adopted from the publication [59]. First of all, the substrates were treated with concentrated hydrochloric acid and methanol mixed at a volume ratio of 1:1 at 60 °C for 30 min, rinsed with distilled water and acetone and dried. After this, the substrate surface was hydroxylated by concentrated sulfuric acid at 60 °C for 30 min. After rinsing and drying, the substrates were immersed in a 2.5 g/L aqueous polyethylenimine solution acidified to pH ≈ 9 for 20 min to obtain a positively charged film for further assembling negatively charged perovskite nanosheets. Then, the substrates were immersed in the diluted suspensions with nanosheet concentrations of approximately 25 mg/L and pH ≈ 11 for 20 min. Finally, the substrates with the deposited nanosheets were rinsed with distilled water and dried at 250 °C for 30 min.

### 2.6. Investigation of the Nanosheet Suspension Stability at Various pH

The approximate range of pH values corresponding to the stability of the nanosheet suspensions towards fast coagulation and sedimentation was determined visually via their acid-base titration. For this, under continuous stirring, the titrant aqueous solution (KOH or HNO_3_) was added dropwise in the amount necessary to change the pH by 0.25 units. When the pH value was stabilized, the solution was transmitted through by a red laser beam (633 nm) and the heterogeneity of the scattered light (Tyndall effect) was evaluated visually. Then, the titration was continued. For the suspensions within the stability range, taken in pH increments of 0.5 units, ζ-potentials were also measured.

### 2.7. Instrumentation

#### 2.7.1. XRD

Powder X-ray diffraction (XRD) analysis of the samples was performed on a Rigaku Miniflex II benchtop Röntgen diffractometer (Tokyo, Japan) using CuK_α_ radiation, an angle range 2θ = 3–60° and a scanning rate of 10°/min. Tetragonal lattice parameters were calculated on the basis of all the reflections observed using DiffracPlus Topas 4.2 software (Bruker, Karlsruhe, Germany). Powder diffraction files (PDF) were taken from The International Centre for Diffraction Data (ICDD) using Rigaku PDXL 2.0 software.

#### 2.7.2. ICP-AES

Nanosheet concentrations in the standard suspensions used for building spectrophotometric calibration plots were determined by inductively coupled plasma atomic emission spectroscopy (ICP-AES) on a Shimadzu ICPE-9000 spectrometer (Kyoto, Japan) after preliminary dissolution of the suspended particles in concentrated hydrochloric acid.

#### 2.7.3. UV-Vis Spectrophotometry

Nanosheet concentrations in other suspensions obtained were controlled via their spectrophotometric analysis on a Thermo Scientific Genesys 10S UV-Vis spectrophotometer (Waltham, MA, USA) using the calibration plots built previously. The spectra were recorded in the range of 190–1100 nm.

#### 2.7.4. DLS

Particle size distribution in nanosheet suspensions, as well as their ζ-potentials, were estimated by the method of dynamic light scattering (DLS) on a Photocor Compact-Z analyzer (Moscow, Russia) using DynaLS 2.0 software (Photocor, Moscow, Russia). The scattered light was detected at an angle of 90° and 20°, respectively.

#### 2.7.5. TEM

Morphology of the nanosheets was studied on a Zeiss Libra 200FE transmission electron microscope (TEM) (Oberkochen, Germany). The main interplanar distances were evaluated based on selected area electron diffraction (SAED) data. Before the deposition on the sample holder, the suspensions were diluted with methanol to achieve the nanosheet concentration of ≈15–20 mg/L.

#### 2.7.6. SEM

Morphology of the bulk precursors for exfoliation and nanosheets deposited on substrates was investigated on a Zeiss Merlin scanning electron microscope (SEM) (Oberkochen, Germany) equipped with a field emission cathode, an electron optics column Gemini II and an oil-free vacuum system.

#### 2.7.7. AFM

Thickness of the nanosheets deposited on substrates was studied using an Integra-Aura atomic force microscope (AFM) (NT-MDT, Zelenograd, Russia) and Gwyddion 2.58 software (Czech Metrology Institute, Jihlava, Czech Republic).

#### 2.7.8. pH Measurement

The pH values of the nanosheet suspensions were controlled using a laboratory pH-meter Mettler Toledo SevenCompact S220 equipped with an InLab Expert Pro-ISM electrode (Greifensee, Switzerland).

## 3. Results and Discussion

### 3.1. Identification of the Precursors for Exfoliation

This research involved the use of two types of precursors for exfoliation: protonated titanates and their amine derivatives produced via the intercalation of *n*-alkylamines with various chain lengths into the interlayer space. All the aforementioned precursors have been synthesized and comprehensively characterized in full accordance with the methods from our previous reports [27,57]. Here, we briefly highlight only the key points of their characterization.

As follows from the results of powder XRD analysis (Appendix A), all the protonated titanates and their *n*-alkylamine derivatives obtained represent individual crystalline compounds not containing perceptible impurity phases. The tetragonal lattice parameters of the protonated precursors (Appendix A) are in good consistency with the literature values [73,74]. The intercalation of *n*-alkylamines is accompanied by a characteristic low-angle shift of the (00x) reflections in the XRD patterns (Appendix A) and results in a corresponding increase in the values of the *c* parameter and interlayer distance *d*, being generally proportional to the *n*-alkylamine length. The alkyl chains form a paraffine-like bilayer with a tilt angle to the perovskite slab of 71–77° [27,57]. Thus, the layered titanate structure experiences pronounced interlayer expansion, which should facilitate its further exfoliation into separate perovskite slabs. 

According to the spectroscopic data [27,57], the interlayer *n*-alkylamines exist in a cationic (alkylammonium) form and, thus, are associated electrostatically with interlayer oxygens of perovskite octahedra. The organic content in the samples is approximately 0.35–0.45 amine units per one interlayer proton. Moreover, the *n*-alkylamine derivatives contain perceptible amounts of intercalated water, while the initial protonated titanates are seen to be practically anhydrous compounds (Appendix A).

Both initial protonated titanates and their *n*-alkylamine derivatives have lamellar particles with a relatively large spread of lateral size (0.5–4 μm) and thickness (200–600 nm), which is predominantly a result of high-temperature ceramic synthesis of the alkaline precursors (Appendix A). However, the amine-intercalated derivatives contain a greater particle fraction with sizes not exceeding 1 μm, as well as demonstrate pronounced slits, fissures and other signs of partial crystal delamination, which make these samples promising precursors for exfoliation into nanosheets.

### 3.2. Spectrophotometric Calibration Plots

It was established that general spectrum appearance for a specific titanate remains similar under all the exfoliation conditions tested in this research, and optical density depends linearly on the nanosheet concentration. This observation indirectly indicates that the optical properties of the nanosheet-containing suspensions and, consequently, morphology features of the nanoparticles do not depend too much on the exfoliation conditions investigated, which allows for the use of the spectrophotometric calibration plots built on the basis of the standard suspensions to control the exfoliation efficiency in other experiments (Figure 2).

Since the UV-vis spectra were recorded without an integrating sphere, the values of optical density observed correspond to the superposition of light scattering and absorption by the dispersed solid phase. Unlike suspensions of bulk non-exfoliated samples [94], those of nanosheets demonstrate practically zero optical density in the visible region (λ > 400 nm), which points to the absence of relatively large particles capable of providing Mie scattering. The ultraviolet range of the spectra used for building calibrations (Figure 2) shows several wide overlapping bands and is individual for n = 1 and n = 3 titanates. When choosing the analytical wavelengths, it was required that they correspond to the band maximum and be sufficiently far from the TBAOH absorption region (λ < 215 nm) to reduce the error associated with baseline subtraction.

### 3.3. Optimization of the Exfoliation Procedure

In this research, the exfoliation efficiency achieved using physical, chemical and combined physical–chemical strategies (Table 2) was evaluated in terms of the nanosheet concentrations in the final suspensions after centrifuging (c, mg/L) and the yields (η, %) from theoretically possible in the case of hypothetical complete exfoliation of the sample (Figure 3, Appendix A). In general, the exfoliation efficiency controlled in this way is predetermined by two key factors: the ability of the chosen precursor to undergo delamination as such and the stability of the resulting nanosheet suspension with respect to centrifuging.

Physical exfoliation of the protonated titanates, consisting in their sonication in water, exhibited very poor efficiency (Figure 3a): the final suspensions contained only trace amounts of the dispersed solid phase (c < 2 mg/L, η < 0.2%). The use of methylamine-intercalated precursors allowed nanosheet yields to be enhanced by 2.5–3.5 times (c < 6 mg/L, η < 0.6%). A more pronounced propensity for exfoliation of the methylamine derivatives may be caused by their greater interlayer distance *d* (Appendix A), facilitating delamination of the layered structure.

Chemical exfoliation of the protonated samples in aqueous TBAOH without sonication (Figure 3a) also proved to be inefficient (c < 3 mg/L, η < 0.3%). However, the methylamine derivatives provided better yields (c < 17 mg/L, η < 1.7%) due to greater accessibility of the expanded interlayer space for TBA^+^ cations and water molecules leading to the structure swelling and delamination.

Physical–chemical exfoliation, combining TBAOH and ultrasonic treatments, has been established to be a powerful technique for producing perovskite nanosheets of the protonated Dion–Jacobson phases. For instance, HB_2_Nb_3_O_10_ (B = Ca, Sr) niobates have been used to yield suspensions with nanosheet concentrations 500 and 950 mg/L, respectively, under the exfoliation conditions taken as default in this research [94]. However, this approach proved to be poorly applicable to the protonated Ruddlesden–Popper titanates under consideration, having given relatively low nanosheet yields (c < 5 mg/L, η < 0.5%) (Figure 3b). The probable reason for this consists in the sharply different chemical activity of the Dion–Jacobson and Ruddlesden–Popper phases with respect to hydration and intercalation reactions. The Dion–Jacobson niobates have lower interlayer charge density, readily undergo interlayer hydration and intercalate directly relatively bulk organic molecules under mild conditions [23,55]. Thanks to this, they are capable of incorporating TBA^+^ cations that facilitate subsequent hydration and delamination upon the ultrasonic treatment. The Ruddlesden–Popper titanates, on the contrary, possess doubled interlayer charge density, usually exist in a low-hydrated or practically anhydrous state and intercalate directly only small molecules, such as methylamine [27]. In view of this, they are unlikely to incorporate TBA^+^ quantitatively, which hinders their liquid-phase exfoliation.

To overcome this obstacle, the *n*-alkylamine derivatives of the titanates were taken as precursors for producing nanosheets (Figure 3b). This approach demonstrated incredibly high efficiency and allowed us to obtain up to 160 times greater nanosheet concentrations and yields in comparison with the aforementioned exfoliation of the protonated titanates under the same conditions: up to 50 mg/L (5.0%) for HLT, 750 mg/L (75%) for HNT, 170 mg/L (17%) for HLT_3_ and 460 mg/L (46%) for HNT_3_. The exfoliation efficiency was revealed to be strongly dependent on the *n*-alkylamine chain length (Figure 3b): it increases drastically when moving from the protonated titanates to their methylamine derivatives, reaches its maximum value for ethylamine derivatives, decreases moderately when going to *n*-propylamine ones and falls sharply upon the further elongation of the *n*-alkylamine chain. Particularly, *n*-hexylamine and *n*-octylamine derivatives undergo exfoliation reluctantly and provide practically the same nanosheet yields as the initial protonated titanates. Apparently, one of the key factors predetermining the propensity for exfoliation of the amine-intercalated precursors is improved availability of their interlayer space for TBA^+^ cations and water molecules. The highest exfoliation efficiency is observed for the derivatives with short-chain *n*-alkylamines (R = Me, Et, Pr). Their molecules, on the one hand, provide a sufficient interlayer space expansion and, on the other hand, are polar enough to facilitate the water incorporation. Despite a greater interlayer distance, the long-chain precursors (R = Hx, Oc) are less favorable for the insertion of water molecules since the latter have to make contact with non-polar alkyl chains. Moreover, these precursors may be stabilized by alkyl–alkyl interactions between the amine chains, preventing the sample from separating into individual perovskite slabs. In addition, the *n*-alkylamines may be partially retained on the nanosheet surface after exfoliation and, thus, affect the suspension’s stability towards aggregation and sedimentation. In particular, an aqueous suspension of nanosheets with adsorbed hydrophilic methylamine is most likely to be more stable than one with adsorbed hydrophobic *n*-octylamine. In view of the highest exfoliation ability, the ethylamine derivatives were chosen as precursors for all the subsequent experiments.

More concentrated suspensions of the titanate nanosheets can be obtained using increased precursor weights (with a proportionately enlarged TBAOH concentration) (Figure 3c). Particularly, a three-fold increase in the sample weight provides a two-fold increase in the nanosheet concentration, and a five-fold increase in the weight results in a three-fold increase in the concentration. In this way, the latter reaches impressive values of up to 2100 mg/L for HNT and 1200 mg/L for HNT_3_. However, the nanosheet yield from the theoretically possible decreases with the increase in the precursor loading due to the limited stability of overly concentrated suspensions.

The precursor and TBAOH amounts in this research were chosen in such a way as to provide a 1:1 ratio of TBA^+^ cations to interlayer vertices of perovskite octahedra. This ratio was found to be optimal since both a two-fold deficiency and a two-fold excess of TBAOH lead to a reduction in resulting nanosheet concentrations and yields (Figure 3d).

It was shown that the exfoliation efficiency increases by two times when changing the sonication power from 25% to 50% from the nominal value (200 W). However, the further power growth hardly affects the nanosheet yield (Figure 3e), which indicates that it is limited by other factors. Moreover, the use of excessively high ultrasound power may adversely influence the nanoparticle morphology, lead to the fast suspension overheating and premature sonotrode corrosion resulting in the suspension contamination with its material. In view of this, the 50% sonication power was considered an optimal value.

It was also established that both stages of the ultrasonic treatment (after preparing the initial suspension and after its stirring for a predetermined time) are important for the efficient liquid-phase exfoliation. When the first stage is skipped, the final nanosheet yield is, on average, 15% less than in the standard experiment with two 5 min ultrasound treatments. Without the second sonication stage, the yield drops by about 25%. It is reasonable to assume that the first stage allows one to «whip up» the layered matrix and facilitate the penetration of water and TBA^+^ cations into the interlayer space. During subsequent stirring, the bulk sample continues swelling and the second sonication, apparently, is intended to promote quantitative delamination of the swollen material. However, performing the sonication for too long turned out to be unnecessary for achieving relatively high exfoliation completeness in the case of the titanates under consideration (Figure 3f). Particularly, an increase in the sonication time from 1 to 5 min is indeed justified and leads to a 20–30% greater nanosheet yield. At the same time, its further elongation up to 10 min improves the exfoliation efficiency by no more than 5%. 

Meanwhile, the duration of suspension stirring between ultrasonic treatments also influences the final nanosheet concentration, especially in the case of HLT and HLT_3_ titanates (Figure 3g). For instance, 7 d exfoliation experiments result in 4–5 times higher nanosheet yields than 1 d ones. Although the subsequent experiment elongation of up to 21 d has a weaker effect on the concentrations, it allows achieving the greatest nanosheet yields obtained in this research: 270 mg/L (27%) for HLT, 950 mg/L (95%) for HNT, 810 mg/L (81%) for HLT_3_ and 920 mg/L (92%) for HNT_3_. Prolonged stirring, probably, is of high importance for greater completeness of TBA^+^ intercalation and hydration reactions since they could proceed slowly due to steric limitations.

Strictly speaking, the similar nanosheet concentrations could be reached in the experiments with standard 1 d of stirring but milder centrifuging conditions (Figure 3h). However, the use of low separation factors preserves relatively bulky particle fractions in the final suspensions, in addition to the desired nanosheets, which was observed from high suspension turbidity and non-zero optical density in the visible region of UV-vis spectra caused by intense Mie scattering. The suspensions obtained in this way may be of interest for specific applications, although, strictly speaking, they are not the suspensions of nanosheets as such.

In view of the above, the optimal conditions of physical–chemical exfoliation (in terms of the nanosheet yield) imply two-stage sonication (50% power, 5 min each one) of ethylamine-intercalated titanates (1 g/L) in 0.004 M TBAOH with intermediate stirring for 7 d and subsequent centrifuging with a separation factor of 1000 for 1 h. If greater nanosheet concentrations (1–2 g/L) are required, the precursor loading and corresponding TBAOH amount should be increased, although it will lead to a decrease in the exfoliation yield.

### 3.4. Investigation of the Nanosheet Morphology

Subsequent studies were conducted using the suspensions prepared via the exfoliation of ethylamine-intercalated titanates under the aforementioned default conditions (standard suspensions).

Particle hydrodynamic radii R_h_ in the standard suspensions of nanosheets were estimated using the DLS method (Figure 4). The particle size distribution for HLT shows a maximum at approximately 90 nm and is significantly different from the distributions for other titanates due to its relative narrowness: it covers the range of 65–125 nm and has a width at half maximum of 40 nm. The distribution for HNT with a maximum at 125 nm is much wider; it falls in the region of 50–250 nm, has a width at half maximum of about 120 nm and exhibits pronounced asymmetry due to the shoulder extending to the area of large radii. The distributions for HLT_3_ and HNT_3_ are practically identical: they correspond to the predominant particle size of 120 nm, cover the range of approximately 50–210 nm and possess a width at half maximum of 100 nm. All the size distributions also exhibit an additional maximum at 15–30 nm that might relate to finer particle fractions. The DLS data presented give only an upper estimate of the actual particle sizes and do not allow their rigorous assessment since the distributions were calculated assuming a spherical shape of the dispersed particles, which is very different from that of the nanosheets under consideration. However, the DLS results are consistent with the data of UV-vis spectrophotometry (Figure 2): HNT, HLT_3_ and HNT_3_ titanates show practically the same particle size distributions and give similar spectra with well-distinguishable maxima around 225–230 nm, while for the HLT compound the maximum is located at 217 nm. Moreover, the DLS results are also consistent with the exfoliation efficiency (Figure 3): among the four amine-intercalated titanates, HLT was found to exhibit the lowest exfoliation yields in this research.

In accordance with TEM images (Figure 5), the main part of the nanosheets obtained has a rectangular shape with lateral dimensions of 30–250 nm, which does not exceed the values estimated by DLS. These sizes are, on average, a bit smaller than those reported earlier for the nanosheets of other layered perovskite-like oxides [97,98,99,100,101,102]. This difference might be a result of unequal exfoliation conditions. Particularly, 1 h centrifuging with a separation factor of 1000 could explain the presence of only fine particle fractions in the final suspensions. Analysis of the titanate nanosheets by means of SAED (Figure 6) allowed for the identification of the main interplanar distances in their structure. The brightest SAED rings correspond to the spacings of approximately 2.62–2.74 Å, 1.86–1.94 Å and 1.31–1.37 Å. Taking into account XRD data for the parent titanates (PDF cards 01-075-2761, 01-076-3080, 00-048-0983 and 01-087-0479), these spacings should probably be attributed to the families of crystallographic planes with Miller indices (110), (020) and (220), respectively. Thus, the structure of the perovskite slabs is seen to be preserved during the exfoliation procedure.

SEM images of the exfoliated titanates (Appendix A) allow for the observation of individual nanosheets deposited on silicon substrates. It is clearly seen that, in addition to relatively large nanosheets with lateral dimensions of about 150–300 nm, all the suspensions contain smaller particle fractions with linear sizes not exceeding 50 nm. The presence of these fractions may explain the aforementioned maxima at 15–30 nm in the particle size distributions determined by means of DLS (Figure 4).

According to AFM data (Figure 7), the predominant height of the deposited nanosheets is 2.0–2.5 nm, which approximately corresponds to the thickness of the titanate monolayer (one perovskite slab). Meanwhile, bilayer nanosheets (4.0–4.5 nm) were detected in significantly smaller quantities.

### 3.5. Investigation of the Nanosheet Suspension Stability at Various pH

The as-prepared suspensions of titanate nanosheets exhibit a pH value of 11.7, which may be too high for use in catalysis, photocatalysis as well as for nanosheet self-assembly with other particles to produce composite nanomaterials. In view of this, the issue of suspension stability at various pH deserves special attention.

The nanosheets in the initial suspensions are characterized by a moderately negative ζ-potential of −23.5 mV for HLT and −26–−27 mV for HNT, HLT_3_ and HNT_3_. The less-negative value for HLT in comparison with the other samples indicates lower aggregative stability of the HLT nanosheet suspensions and is consistent with the reduced exfoliation ability of this titanate. Both acidification and alkalization of the medium leads to a decrease in the absolute value of ζ-potential because of the rising ionic strength and resulting double layer contraction (Figure 8a). Weakening of the electrostatic barrier leads to pronounced particle aggregation that can be easily observed via the illumination of the suspension using a laser beam (Figure 8b). When the nanosheet suspension is stable, the scattered light (a Tyndall effect) remains homogeneous and individual scattering particles are invisible to the naked eye. As the intense nanosheet aggregation begins, separate sparkles become easily noticeable in the scattered light. The concentration of the aggregated particles continues increasing and, when the critical pH is reached, the suspension experiences fast flocculation. The latter is accompanied by a severe increase in turbidity due to the active formation of flakes that begin to settle on the bottom immediately after turning off the magnetic stirrer. The n = 1 titanates were shown to undergo fast flocculation and sedimentation when the pH is lower than 8.0 or higher than 12.5, while for the n = 3 titanates the corresponding values are 7.5 and 12.0, respectively. The ζ-potentials measured at these pHs depend on a specific titanate (Figure 8a) and fall in the range of −16–−10 mV. However, clear signs of the nanosheet agglomeration were already observed at pH ≤ 9.5 when ζ ≥ −15 mV for the n = 1 samples and ζ ≥ −20 mV for the n = 3 ones. In general, the nanosheet suspensions of the titanates under consideration exhibit narrower stability range (≈4.5 pH units) than those of the Dion–Jacobson niobates HB_2_Nb_3_O_10_ (B = Ca, Sr) studied earlier (≈6.5 units) [94].

## 4. Conclusions

The present research has suggested and systematically tested a highly efficient approach to the quantitative liquid-phase exfoliation of the Ruddlesden–Popper phases HLnTiO_4_ and H_2_Ln_2_Ti_3_O_10_ (Ln = La, Nd). Protonated forms of these titanates demonstrate high stability towards delamination into separate perovskite slabs and do not allow for the production of nanosheets with acceptable yields. However, their preliminary modification with *n*-alkylamines has provided up to 160 times greater exfoliation efficiency and has allowed for the achievement of impressive nanosheet concentrations in suspensions up to 2.1 g/L and yields of up to 95% for the first time. The intercalated amines expand the interlayer space and, apparently, facilitate the accommodation of tetrabutylammonium cations and water molecules, which are necessary for the swelling of the layered precursor and its prying apart upon sonication. An issue of high importance is the *n*-alkylamine length: while the samples with short-chain amines readily undergo exfoliation, those with long-chain ones do it reluctantly. Although the ultrasonic treatment is essential for the quantitative exfoliation, too long sonication (≥10 min) proves to be unnecessary for achieving high nanosheet yields for the titanates under study. Meanwhile, the elongation of the suspension stirring from 1 d to 7 d exhibits pronounced beneficial influence on the exfoliation completeness. Moreover, the latter was shown to depend on a specific titanate: Nd-containing precursors in general give more concentrated suspensions than La-containing ones after stirring for a short time (1 d), despite the fact that their protonated and amine-intercalated forms are generally similar [27,57]. While the difference in the exfoliation efficiency between H_2_Ln_2_Ti_3_O_10_ compounds becomes insignificant upon greater stirring durations (≥7 d), the HLaTiO_4_ compound remains quite special in terms of the lower exfoliation degree, particle size distribution and particle ζ-potential. Successful preparation of the titanate nanosheets was confirmed using dynamic light scattering, electron and atomic force microscopy; nanosheet lateral sizes fall in the range of 30–250 nm, and their prevailing thickness is 2–4 nm.

Thus, the Ruddlesden–Popper titanates HLnTiO_4_ and H_2_Ln_2_Ti_3_O_10_ (Ln = La, Nd) can be efficiently exfoliated into nanosheets after preliminary interlayer modification with short-chain alkylamines. The developed exfoliation strategy may be of interest for the high-yield production of perovskite nanosheet-based materials for photocatalytic hydrogen production, environmental remediation and other applications.

## Figures and Tables

**Figure 1 nanomaterials-13-03052-f001:**
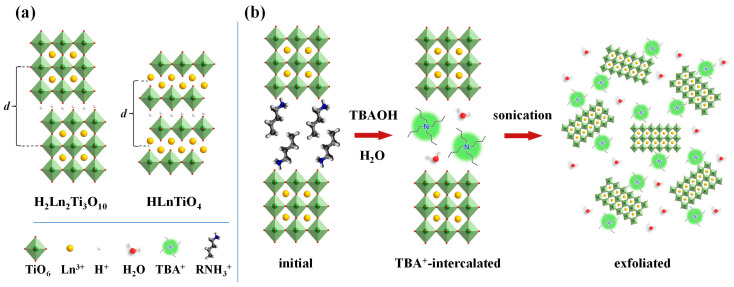
Schematic structure of protonated compounds HLnTiO_4_ and H_2_Ln_2_Ti_3_O_10_ (**a**), physical–chemical exfoliation of amine-intercalated H_2_Ln_2_Ti_3_O_10_ into nanosheets (**b**).

**Figure 2 nanomaterials-13-03052-f002:**
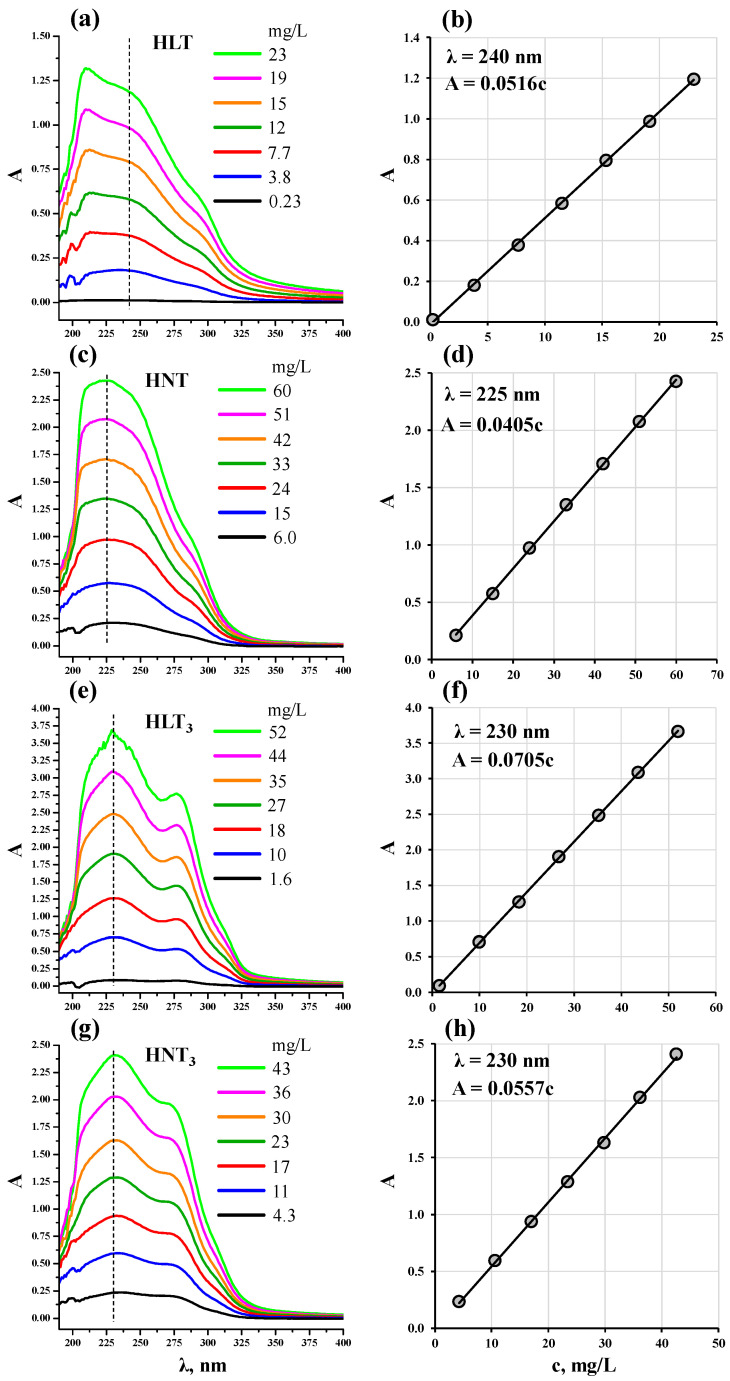
UV-vis spectra of the nanosheet suspensions with various dilutions and corresponding calibration plots for express measurement of HLT (**a**,**b**), HNT (**c**,**d**), HLT_3_ (**e**,**f**) and HNT_3_ (**g**,**h**) nanosheet concentrations.

**Figure 3 nanomaterials-13-03052-f003:**
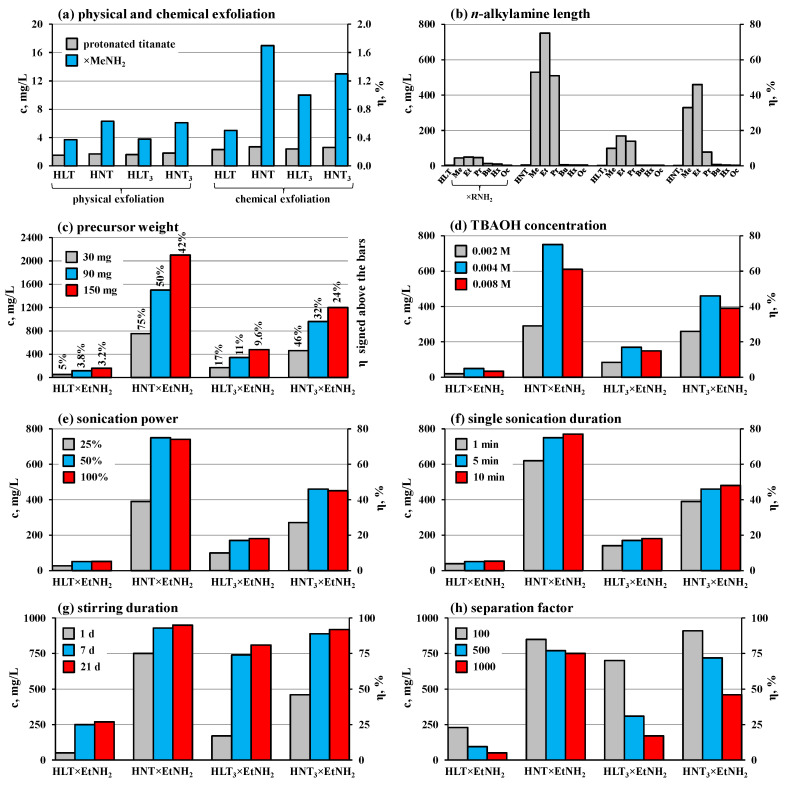
Influence of exfoliation approach (**a**), interlayer *n*-alkylamine length (**b**), precursor weight (**c**), TBAOH concentration (**d**), sonication power (**e**), sonication duration (**f**), stirring duration (**g**) and centrifuge separation factor (**h**) on final nanosheet concentration and yield.

**Figure 4 nanomaterials-13-03052-f004:**
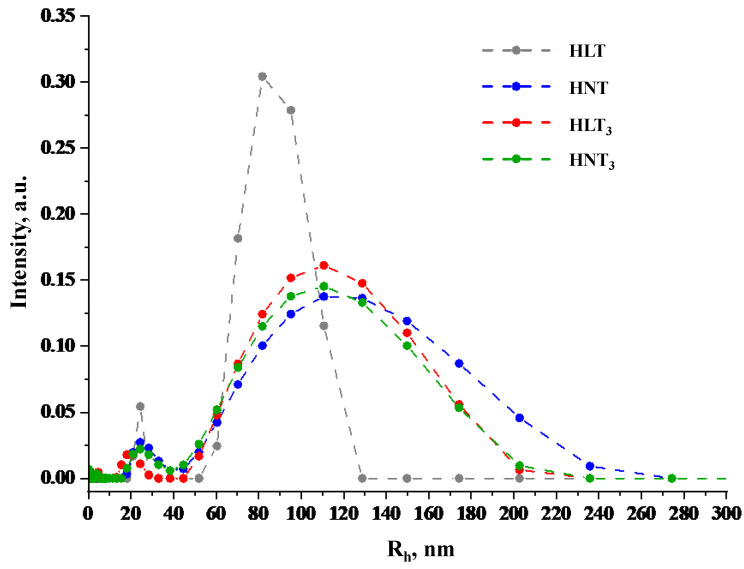
Particle size distributions in the nanosheet suspensions obtained by DLS.

**Figure 5 nanomaterials-13-03052-f005:**
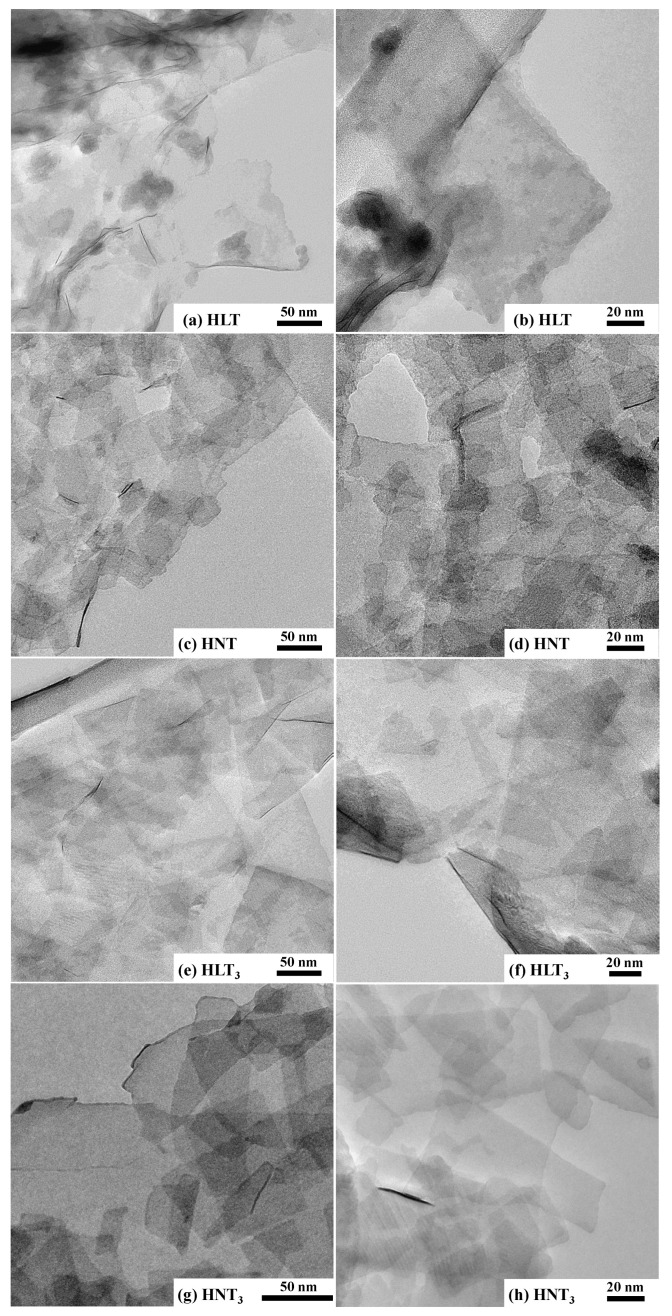
TEM images of HLT (**a**,**b**), HNT (**c**,**d**), HLT_3_ (**e**,**f**) and HNT_3_ (**g**,**h**) nanosheets.

**Figure 6 nanomaterials-13-03052-f006:**
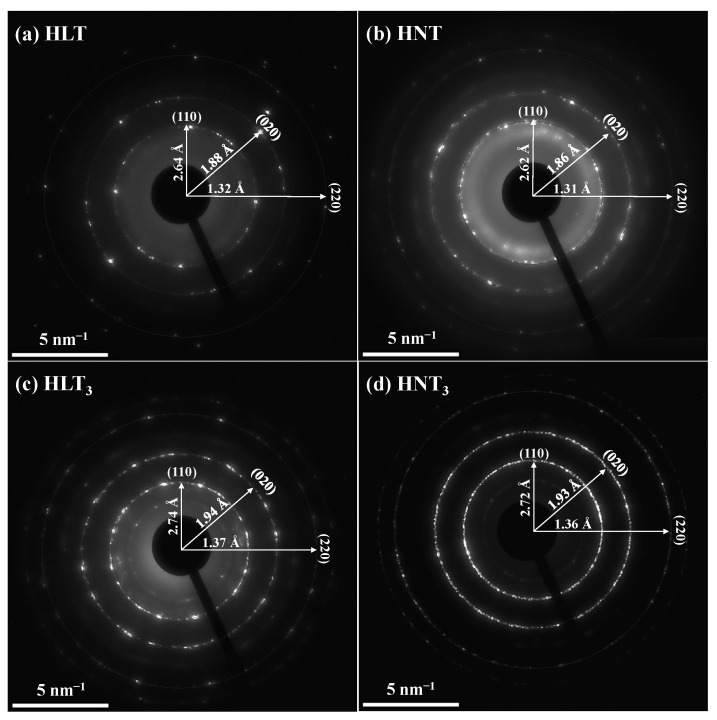
SAED patterns of HLT (**a**), HNT (**b**), HLT_3_ (**c**) and HNT_3_ (**d**) nanosheets.

**Figure 7 nanomaterials-13-03052-f007:**
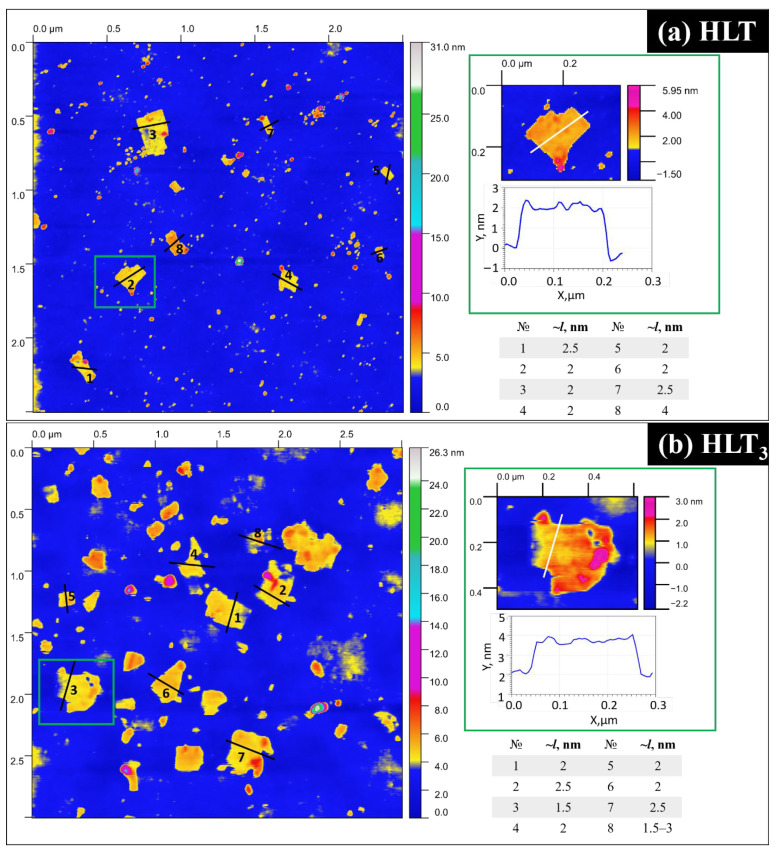
AFM profiles for HLT (**a**) and HLT_3_ (**b**) nanosheets deposited on silicon substrates.

**Figure 8 nanomaterials-13-03052-f008:**
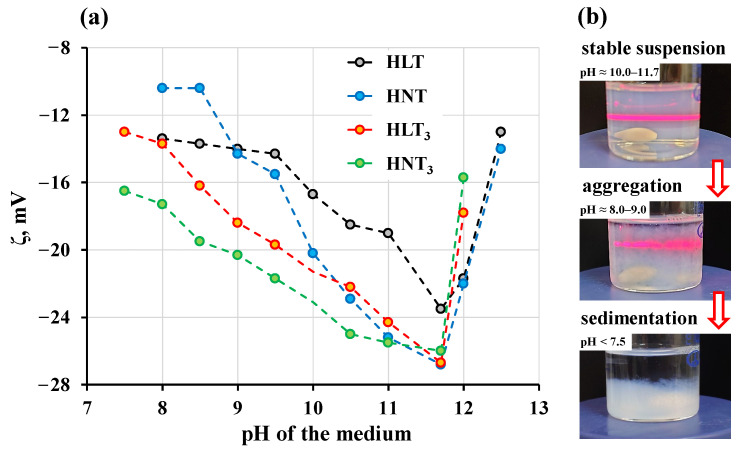
Dependence of ζ-potential on pH of the nanosheet suspensions (**a**) and particle flocculation upon pH shifts (**b**).

**Table 1 nanomaterials-13-03052-t001:** Conditions for synthesis of *n*-alkylamine derivatives.

R	Precursors	Amine Concentration, %	T, °C	D, d	Solvent for Flushing
Me	HLT (HNT), HLT_3_ (HNT_3_)	38 (in water)	60	10	acetone
Et	HLT×MeNH_2_ (HNT×MeNH_2_), HLT_3_×MeNH_2_ (HNT_3_×MeNH_2_)	70 (in water)	25	1	
Pr		90 (in water)			
Bu					
Hx		30 (in *n*-heptane)	60	7	*n*-hexane
Oc					

**Table 2 nanomaterials-13-03052-t002:** Investigated conditions for exfoliation of the titanates into nanosheets. Here HLT and HLT_3_ refer to both La- and Nd-containing samples.

Experiment Series	Precursor	Weight, mg	TBAOH, M	Sonication		Stirring, d	Separation Factor
				Power, %	Duration, min		
Physical exfoliation	HLT, HLT_3_	30	-	100	5	1	1000
	×MeNH_2_						
Chemical exfoliation	HLT, HLT_3_	30	0.004	−	−	1	1000
	×MeNH_2_						
Physical–chemical exfoliation	HLT, HLT_3_	30	0.004	50	5	1	1000
	×MeNH_2_						
	×EtNH_2_						
	×PrNH_2_						
	×BuNH_2_						
	×HxNH_2_						
	×OcNH_2_						
Precursor weight	×EtNH_2_	30	0.004	50	5	1	1000
		90	0.012				
		150	0.02				
TBAOH concentration		30	0.002	50	5	1	1000
			0.004				
			0.008				
Sonication power		30	0.004	25	5	1	1000
				50			
				100			
Sonication duration		30	0.004	50	1	1	1000
					5		
					10		
Stirring duration		30	0.004	50	5	1	1000
						7	
						21	
Separation factor		30	0.004	50	5	1	100
							500
							1000

## Data Availability

The data presented in this research are available in the article body and Appendix A.

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
