# Peer review of "Highly Efficient Liquid-Phase Exfoliation of Layered Perovskite-like Titanates HLnTiO4 and H2Ln2Ti3O10 (Ln = La, Nd) into Nanosheets"

_nanomaterials, 2023, doi:10.3390/nano13233052_

Round 1
Reviewer 1 Report
Comments and Suggestions for Authors
S. A. Kurnosenko et al synthesized perovskite type titanate nanosheets using physical-chemical methods. HLnTiO4 and H2Ln2Ti3O10 were used as precursors, and n-alkalines were intercalated to enhance their layer distance. The physical-chemical process for exfoliation of the precursors achieved a high yield efficiency of perovskite nanosheets of up to 95%. The Authors characterized the obtained nanosheets using DLS, TEM, SAED, and AFM, which confirmed the perovskite structures and nanosheet size of 150-300 nm dimension and 2.0-2.5 nm thickness.
Recently, oxide nanosheets have been paid attention as 2D materials were enthusiastically investigated. This manuscript provides guideline for the high yield efficiency of perovskite nanosheets. Various conditions were applied for exfoliation of the precursors, and the yield of the nanosheets was compared. In contrast, characterizations were only performed from a structural viewpoint, and the physical properties were hardly investigated in the present study. As a demonstration of the nanosheet formation, this work would be useful for readers who performed similar studies. Before my recommendation, I suggest some revisions to this manuscript. My suggestions are listed as follows.
1) The UV spectra and their analyses were shown in Fig. 2. Firstly, this figure should be moved to the Results and Discussion section. The fitting condition should be reconsidered. The linear relationship between A and c is OK, but another condition of (A, c) = (0, 0) should be added when fitting. This condition is satisfied in principle.
2) In Section 3.2, the Authors optimized exfoliation conditions. Please summarize the best condition at the end of the section, which is quite helpful for readers.
3) In Fig. 6, the SAED patterns were shown for the nanosheet samples. The Authors’ assignment of the d values was strange to me. Why are the ring patterns of “2*(3.7-3.8)Å” far from those of “(3.7-3.8)Å”?
4) On Line 458, the Authors indicated “the predominant height of the deposited nanosheets is 2.0–2.5 nm, which approximately corresponds to the thickness of the titanate monolayer (one perovskite slab). Meanwhile, bilayer nanosheets (4.0–4.5 nm) were detected in significantly smaller quantities.” However, the length of the one perovskite slab is not 2.0–2.5 nm but 2.0–2.5 Å, implying that several perovskite slabs were combined in the nanosheets.
5) In Fig. 8(b), please add the corresponding pH for each solution condition: stable suspension, aggregation, and sedimentation.
Comments on the Quality of English LanguageI do not find any critical issues on the Authors' English.
Author Response
Reviewer 1
- The UV spectra and their analyses were shown in Fig. 2. Firstly, this figure should be moved to the Results and Discussion section. The fitting condition should be reconsidered. The linear relationship between A and c is OK, but another condition of (A, c) = (0, 0) should be added when fitting. This condition is satisfied in principle.
We moved this figure to the Results and Discussion section. We added zero points to all calibration graphs and updated the corresponding equations.
- In Section 3.2, the Authors optimized exfoliation conditions. Please summarize the best condition at the end of the section, which is quite helpful for readers.
We summarized the optimal exfoliation conditions at the end of the section.
- In Fig. 6, the SAED patterns were shown for the nanosheet samples. The Authors’ assignment of the d values was strange to me. Why are the ring patterns of “2*(3.7-3.8)Å” far from those of “(3.7-3.8)Å”?
We completely revised the SAED section as well as matched the spacings found with those from the XRD data. Updated Figure 6 and its discussion are presented in the manuscript.
- On Line 458, the Authors indicated “the predominant height of the deposited nanosheets is 2.0–2.5 nm, which approximately corresponds to the thickness of the titanate monolayer (one perovskite slab). Meanwhile, bilayer nanosheets (4.0–4.5 nm) were detected in significantly smaller quantities.” However, the length of the one perovskite slab is not 2.0–2.5 nm but 2.0–2.5 Å, implying that several perovskite slabs were combined in the nanosheets.
Apparently, the question is not about the length, but about the height of the plates. The minimum possible height of nanoplates after the exfoliation of a layered oxide is achieved in the case of delamination into separate perovskite monolayers. The thickness of such a monolayer should be slightly higher than the interlayer distance of the corresponding layered oxides ~12 Å for HLT and ~13.5 Å for HLT3 (Table S1).
- In Fig. 8(b), please add the corresponding pH for each solution condition: stable suspension, aggregation, and sedimentation.
We added pH values to the Figure 8(b).
Reviewer 2 Report
Comments and Suggestions for Authors
In this paper, Sergei A. Kurnosenko et al., provided a systematic exploration of the high-yield exfoliation of Ruddlesden-Popper titanates into nanosheets. The combined pre-intercalation of n-alkylamines and subsequent intercalation of tetrabutylammonium hydroxide leads to efficient exfoliation efficacy with a high nanosheet concentration around 2.1 g/L and 95% yield. Overall, this work is rationally designed, and the paper is also well written, some key experimental factors are comprehensively discussed, and I think this paper is publishable after some minor revisions:
1. Firstly, I would like to say that the authors did a very careful experimental design, most of the key experimental factors including n-alkylamine size, sonication duration, sonication power, and so forth have been comprehensively conducted and discussed. Some minor concerns related to experimental methods shall include:
i) Have the authors considered the size effect of tetra-alkyl ammonium besides TBAOH? The size of ammonium cation also reveals a significant effect for the exfoliation result of many layered materials like black phosphorus (DOI: 10.1021/acs.chemmater.8b00521), transition metal dichalcogenide (DOI: 10.1038/s41563-020-00831-1), and etc. Could the authors clarify this point?
ii) Line 384-397, the authors mentioned that sonication duration revealed a relatively week effect on the exfoliation completeness, this is a bit strange to me, after all, elongation of sonication duration could generally lead to severe breakage of in-plane covalent bond (DOI: 10.1002/anie.201409400), thus yielding small sized nanoparticles or quantum dots, have the authors consider this factor?
2. Perovskite-like crystal trend to suffer from severe stability issue in 2D thickness (10.1038/s41578-020-0185-1), have the authors considered the stability issue in exfoliated dispersion, any interpretation for the chosen of water as dispersing solvent?
3. The image resolution shall be further improved, some details are hard to read in Figure 3.
4. The color range of AFM image in Figure 7 shall be modulated for clarity, the authors shall decrease the maximum height or decrease the minimum height of color-bar in the left section of Figure 7.
Author Response
Reviewer 2
1.1. Have the authors considered the size effect of tetra-alkyl ammonium besides TBAOH? The size of ammonium cation also reveals a significant effect for the exfoliation result of many layered materials like black phosphorus (DOI: 10.1021/acs.chemmater.8b00521), transition metal dichalcogenide (DOI: 10.1038/s41563-020-00831-1), and etc. Could the authors clarify this point?
The use of tetrabutylammonium hydroxide as an exfoliation agent is a standard practice for this class of compounds. Among other quaternary ammonium bases (usually compared TMA+, TEA+, TBA+), TBA+ shows the best swelling and exfoliation rates (for example, DOI 10.1016/j.jcis.2012.09.079,
10.1021/ja501587y). For this reason, we decided to focus only on the use of TBAOH.
1.2. Line 384-397, the authors mentioned that sonication duration revealed a relatively week effect on the exfoliation completeness, this is a bit strange to me, after all, elongation of sonication duration could generally lead to severe breakage of in-plane covalent bond (DOI: 10.1002/anie.201409400), thus yielding small sized nanoparticles or quantum dots, have the authors consider this factor?
Saying this, we meant that long sonication (≥ 10 min) turned out to be unnecessary for achieving high enough concentrations of the titanate nanosheets. This follows from the experimental results: elongation of each sonication stage from 5 to 10 min led to an increase in the yield by only a few percent. However, the difference between 1 min and 5 min sonication was more pronounced (20–30%). To avoid misunderstanding, we clarified the wording in manuscript. The use of relatively long ultrasound treatments may result in the formation of smaller nanosheets, although we did not investigate this issue in detail. However, UV-vis spectra of the final suspensions demonstrated practically the same appearance regardless the sonication time (1, 5, 10 min per one stage). This observation may indirectly indicate that particle sizes and morphology did not change radically upon elongation of the ultrasound treatment in our conditions.
- Perovskite-like crystal trend to suffer from severe stability issue in 2D thickness (10.1038/s41578-020-0185-1), have the authors considered the stability issue in exfoliated dispersion, any interpretation for the chosen of water as dispersing solvent?
In this work, the stability of the resulting colloidal solutions over time was not studied. This is certainly an important characteristic, but in many cases of practical use of such suspensions long-term stability is not required.
The choice of water as a dispersing medium (in addition to the obvious availability, non-toxicity and relative ease of separation) was due to the need to use a solvent in which swelling of intercalates with tetrabutyammonium and other amines can occur, which is a necessary criterion for subsequent exfoliation.
- The image resolution shall be further improved, some details are hard to read in Figure 3.
We improved the image resolution.
- The color range of AFM image in Figure 7 shall be modulated for clarity, the authors shall decrease the maximum height or decrease the minimum height of color-bar in the left section of Figure 7.
We improved the image.
Round 2
Reviewer 1 Report
Comments and Suggestions for Authors
I have read the Authors reply to my comments. I found that the Authors misunderstood my first comment. Please remove the data at the origin since the Authors did not experimentally obtain the data. My suggestion is to add a restraint condition (A, c) = (0, 0) when fitting. In principle, absorption spectra always satisfy this condition. Thus, the fitting curves are lines with the formula A = kc, where k is a constant depending on the nanosheet material.
Author Response
Thank you for the comment. We agree that in this case the fitting curves with the formula A = kc is a reasonable choice. We have made the necessary changes
Round 3
Reviewer 1 Report
Comments and Suggestions for Authors
I agree to the Authors revision. I can recommend this manuscript for publication in Nanomaterials.